# TAS2R38 Genotype Does Not Affect SARS-CoV-2 Infection in Primary Ciliary Dyskinesia

**DOI:** 10.3390/ijms25168635

**Published:** 2024-08-08

**Authors:** Gioia Piatti, Giorgia Girotto, Maria Pina Concas, Leonardo Braga, Umberto Ambrosetti, Mirko Aldè

**Affiliations:** 1Department of Pathophysiology and Transplantation, University of Milan and Unit of Bronchopneumology, Fondazione IRCCS Ca’ Granda Ospedale Maggiore Policlinico, Via Francesco Sforza 35, 20122 Milan, Italy; 2Department of Medicine, Surgery and Health Sciences, University of Trieste and Institute for Maternal and Child Health—IRCCS “Burlo Garofolo”, 20038 Trieste, Italy; giorgia.girotto@burlo.trieste.it; 3Institute for Maternal and Child Health—IRCCS “Burlo Garofolo”, 20038 Trieste, Italy; mariapina.concas@burlo.trieste.it; 4Laboratory of Healthcare Research & Pharmacoepidemiology, Department of Statistics and Quantitative Methods, University of Milano-Bicocca, 20126 Milan, Italy; l.braga1@campus.unimib.it; 5Department of Clinical Sciences and Community Health, University of Milan and Division of Otolaryngology, Fondazione IRCCS Ca’ Granda, Ospedale Maggiore Policlinico, Via Francesco Sforza 35, 20122 Milan, Italy; umberto.ambrosetti@unimi.it (U.A.); mirko.alde@unimi.it (M.A.)

**Keywords:** bitter taste receptors, *TAS2R38*, primary ciliary dyskinesia, COVID-19, SARS-CoV-2

## Abstract

Several chronic respiratory diseases could be risk factors for acquiring SARS-CoV-2 infection: among them, Primary Ciliary Dyskinesia (PCD) is a rare (about 1:10.000) inherited ciliopathy (MIM 242650) characterized by recurrent upper and lower respiratory tract infections due to a dysfunction of the respiratory cilia. In this study, we aimed to investigate whether PCD subjects are more susceptible to infection by SARS-CoV-2 and whether some polymorphisms of the *TAS2R38* bitter taste receptor correlate with an increased prevalence of SARS-CoV-2 infection and severity of symptoms. Patients answered several questions about possible SARS-CoV-2 infection, experienced symptoms, and vaccinations; in the case of infection, they also filled out a SNOT-22 questionnaire and ARTIQ. Forty PCD adult patients (mean age, 36.6 ± 16.7 years; 23 females, 17 males) participated in this study, out of which 30% had tested positive for COVID-19 during the last four years; most of them reported a mildly symptomatic disease. We found no differences in age or sex, but a statistically significant difference (*p* = 0.03) was observed in body mass index (BMI), which was higher in the COVID-acquired group (23.2 ± 3.3 vs. 20.1 ± 4.1 kg/m^2^). Genotyping for *TAS2R38* polymorphisms showed a prevalence of 28.6% PAV/PAV, 48.6% PAV/AVI, and 22.8% AVI/AVI individuals in our cohort. In contrast to our hypothesis, we did not observe a protective role of the PAV allele towards SARS-CoV-2 infection. Conclusions: Our findings suggest that subjects with PCD may not be at increased risk of severe outcomes from COVID-19 and the *TAS2R38* bitter taste receptor genotype does not affect SARS-CoV-2 infection.

## 1. Introduction

Since its identification at the end of 2019, the severe acute respiratory syndrome coronavirus strain 2 (SARS-CoV-2) causing coronavirus disease 2019 (COVID-19) has rapidly spread around the world, causing a pandemic. The spike protein (S) protruding from the envelope of the virus, and resulting in its characteristic crown-like appearance, binds to specific receptors on the host cell surface, such as angiotensin-converting enzyme 2 (ACE 2); these receptors are particularly numerous on airway ciliated cells [1].

Several studies [2,3,4] have highlighted the role of the extra-oral bitter taste receptors (TAS2Rs) in the regulation of the respiratory tract immune responses against infections, especially by Gram-negative bacteria; in contrast, their role against viruses and, particularly, SARS-CoV-2 is less clear. TAS2Rs are a family of G-protein-coupled receptors comprising ~25 isoforms that, apart from their role in bitter taste sensation on the taste buds of the tongue, are largely expressed in airway ciliated epithelia and solitary chemosensory cells and on all vital organs of the human body, where they play a crucial role in regulating the synthesis of anti-microbial peptides and in reducing airway inflammation. Thus, TAS2Rs could potentially modulate SARS-CoV-2 infection. Moreover, gustatory dysfunction is one of the clinical hallmarks in COVID-19 patients; this raises the possibility that TAS2R G protein, gustducin, in the upper and lower respiratory tract could act as a gateway for the SARS-CoV-2 infiltration similar to the known ACE 2 receptor [5].

The most studied bitter taste receptor is TAS2R38, which is encoded by the *TAS2R38* gene in the human genome and expressed as two predominant high-frequency haplotypes, determined by two Single Nucleotide Polymorphisms (SNPs): the functional variant PAV and the non-functional variant AVI. The functional TAS2R38 contains proline (P), alanine (A), and valine (V) residues at positions 49, 262, and 296, respectively, while the non-functional TAS2R38 contains alanine (A), valine (V), and isoleucine (I) in the same positions. These polymorphisms influence the individual sensitivity to the bitter taste; PAV/PAV-taster individuals account for approximately 25% of the adult population, PAV/AVI tasters for approximately 50%, and nontasters (AVI/AVI) for approximately 25% [6,7]. 

In vitro, TAS2R38 activation stimulates the release of nitric oxide (NO) with biocidal activity; in fact, this molecule and its derivatives cause a reduction in viral RNA production interfering with the early steps of viral replication and of viral protein synthesis by inhibiting the docking of the spike protein onto ACE 2 [8]. The activation of extraoral TAS2Rs promotes a release of NO from epithelial ciliated cells and increases their ciliary beat frequency, thereby facilitating the clearance of invading pathogens.

Only a few clinical studies address the relationship between *TAS2R38* polymorphisms and SARS-CoV-2 infection, reporting controversial results. Barham at al. [9] showed that some genotypes of the TAS2R38 bitter taste receptor could be associated with the clinical course of patients exposed to SARS-CoV-2: in particular, AVI/AVI individuals were found to be more frequently positive for SARS-CoV-2, to tend to be hospitalized once infected, and to be symptomatic for a longer duration than PAV/AVI and PAV/PAV, suggesting enhanced innate immune protection against SARS-CoV-2 in the latter two. Parsa et al. [10] also reported that the *TAS2R38* PAV allele rather than AVI is associated with lower COVID-19 mortality. On the other hand, by examining subjects who underwent *TAS2R38* genotyping and became infected by SARS-CoV-2 with different severities of illness, Risso et al. [11] failed to find any significant relationship between the different genotypes and presence/severity of SARS-CoV-2 infection. Furthermore, by studying one-hundred and ninety-six adult patients affected by mild-to-moderate COVID-19, Santin et al. [12] did not identify any significant association between the *TAS2R38* genotype and the presence/absence of symptoms to SARS-CoV-2 infection. 

Several chronic respiratory diseases such as chronic obstructive pulmonary disease (COPD), bronchiectasis, and idiopathic pulmonary fibrosis (IPF) could be risk factors for acquiring SARS-CoV-2 infection and are often associated with poor outcomes of COVID-19 [13,14]: among them, Primary Ciliary Dyskinesia (PCD) is a rare (about 1:10,000) inherited ciliopathy (MIM 242650) characterized by recurrent upper and lower respiratory tract infections due to abnormal function of the cilia; in about 50% of cases, embryonal dysfunction of the primary cilia leads to organ laterality defects. Symptoms such as chronic nasal discharge and wet cough are typical in these patients and usually begins early and persist during the entire life; the respiratory disease tends to progress because recurrent lung infections and inflammation lead to bronchiectasis and a progressive deterioration in lung function. 

In a cohort of PCD subjects that are affected by chronic respiratory disease, we aimed to investigate (1) whether they are more susceptible to SARS-CoV-2 infection and (2) whether the polymorphisms of the TAS2R38 bitter taste receptor correlate with a greater prevalence of SARS-CoV-2 infection and symptom severity. We hypothesized that the functional PAV allele may act as a protective factor towards SARS-CoV-2 infection and symptom severity, while the AVI allele may represent a risk factor. Up to now, the association of *TAS2R38* polymorphisms with the PCD phenotype has never been tested and reported. In this study, for the first time, we look at the distribution of *TAS2R38* genotypes in a cohort of PCD patients and verify the possible correlations between these genotypes and clinical features.

## 2. Results

A total of 40 PCD patients, mainly adults, participated in this study. The median age was 34 years old (range: 10 to 68 years) and 57.5% (23/40) were females. The main features of patients included in this study are listed in Table 1. 

As expected, bronchiectasis and chronic rhinosinusitis were very frequently reported in PCD patients; the main comorbidities were bronchial asthma and allergies.

A total of 12 out of 40 patients, corresponding to 30% of the study population, tested positive for COVID-19 in the course of the last four years; 1 patient tested positive for COVID-19 in two occasions. Most patients experienced a mild symptomatic disease (mild fever and/or cough), only one had an olfaction/taste reduction during infection for a short time, none had pneumonia, and none required hospitalization in intensive care or died. All patients reported being vaccinated against SARS-CoV-2, out of which all but one had received at least 3 shots of a vaccine, with 6 out of 40 (15%) reporting minor side-effects such as arm pain, fever, tiredness, and headache.

When we compared the clinical features of PCD patients who had been infected by COVID-19 to those who had not, we found no differences in age or sex, but a statistically significant difference (*p* = 0.03) was observed in BMI that was higher in the COVID-acquired group (23.2 ± 3.3 vs. 20.1 ± 4.1 Kg/m^2^). The clinical features of PCD patients belonging to the two groups are reported in Table 2.

During the acute phase of COVID-19, the most common symptoms reported by the participants on the ARTIQ were asthenia (83.3%), fever (66.6%), running nose (58.3%), increased cough (41.7%), muscle or joint pain (41.7%), headache (25%), and shortness of breath (25%). The prevalence of reported symptoms in PCD COVID-acquired patients is available as a Appendix A. 

A total of 35 patients underwent genotyping for *TAS2R38* polymorphisms: the prevalence of PAV/PAV individuals was 28.6%, while those of PAV/AVI and AVI/AVI were 48.6% and 22.8%, respectively. The binomial logistic regression analysis failed to show a significant association between COVID-19 infection and *TAS2R38* haplotypes: PAV/PAV vs. AVI/AVI (estimation: 0.25; *p* = 0.81, positive association); PAV/PAV vs. PAV/AVI (estimation: 0.33; *p* = 0.71, positive association); and PAV/AVI vs. AVI/AVI (estimation: −0.08; *p* = 0.94, negative association). Table 3 reports the clinical features of SARS-CoV-2 infection in PCD patients according to *TAS2R38* polymorphisms. 

## 3. Discussion

This is a retrospective study on a cohort of PCD subjects to evaluate the prevalence of COVID-19 in the last four years and a possible correlation between some clinical features of COVID-19 and polymorphisms of the taste bitter receptor TAS2R38.

PCD, similar to other chronic respiratory diseases, may represent a risk factor for SARS-CoV-2 infection; nevertheless, while some studies [15,16,17] have demonstrated a higher risk of intensive care need and mortality in subjects affected by chronic pulmonary obstructive disease (COPD) and cystic fibrosis (CF), SARS-CoV-2 infection in people with PCD seems to be neither frequent nor particularly severe [18].

In our study, the incidence rate of SARS-CoV-2 infection in PCD resulted in 7.5 per 100 persons-years, which is similar to that observed in the general population and comparable to that reported by Pedersen et al. [19]. These authors, by performing a large longitudinal online survey on health and quality of life in 728 PCD subjects (COVID-PCD) during the pandemic from May 2020 to May 2022, reported a low incidence of SARS-CoV-2 (9 per 100 persons-years) and an overall mild severity of disease in these subjects. Furthermore, in the same study, the incidence of SARS-CoV-2 infection was reported highest in adults aged ≥50 years, similar to what occurs in the general population, in which the severity of COVID-19 is strongly associated with age, and most hospitalizations involve people aged ≥70 years. 

We did not find differences in age or sex, but our group did not include elderly people (maximum age: 68 y). 

It was also reported that only 3.4% of the pediatric population have a SARS-CoV-2 infection, because children are more often asymptomatic and SARS-CoV-2 infections thus remain largely undetected [20]; our sample included mainly adult patients (only six patients were of pediatric age) and, therefore, it is not possible to draw any strong conclusions in this regard. 

Interestingly, BMI was higher in our PCD patients who experienced COVID-19 in comparison to those who did not, in agreement with a previous report [21]. 

Consistently with studies by Pedersen et al. [19,22], most PCD patients experienced a mild severity disease (mild fever and/or cough) and nobody was hospitalized or treated in intensive care unit or died.

A collateral study of COVID-PCD [23] on facemask usage among people with PCD during the COVID-19 pandemic showed that these subjects carefully protected themselves against infection by avoiding crowded places and wearing a facemask in public. Most participants in our study also agreed with the notion that facemasks are effective in preventing the transmission of SARS-CoV-2 and they have widely used this prevention measure. According to these authors [23], the low incidence of SARS-CoV-2 infection was probably attributable to the marked care by PCD subjects at reducing social contact, wearing masks in public, and getting vaccinated against COVID-19. Indeed, adults with PCD consider themselves to be at high risk and therefore are particularly careful at protecting themselves. 

Another aspect of COVID-19 in PCD concerns the vaccines: all patients included in our study were vaccinated against SARS-CoV-2 (all but one had received at least three vaccinations) and only a minority of them (15%) reported minor adverse effects after vaccination. This is in line with what was reported by Pedersen et al. [24] who, in PCD subjects, found high rates of vaccine uptake (96%) and no severe side-effects; only mild side-effects were experienced, such as redness, swelling, or pain around the injection site (60% of participants), and participants reported side-effects more often after the second than the first vaccine. These side-effects are the same ones observed in the general population. Furthermore, these mild symptoms were more often reported by younger than older participants; the authors suggest that this may linked to a stronger immune response to the vaccine in younger people. 

The relationship between *TAS2R38* SNPs and viral infection, particularly that of SARS-CoV-2, has not been previously determined in PCD patients. Our study is the first to investigate the possible correlation between SNPs of *TAS2R38* and COVID-19 in a rare chronic respiratory disease, such as PCD. This disease is characterized by recurrent respiratory infections due to the dysfunction of respiratory cilia that compromises the mucociliary clearance of pathogens.

In a previous work [25], we investigated the relationship between *TAS2R38* polymorphisms in a cohort of PCD patients and found that haplotypes AVI/AVI and PAV/AVI are correlated with some clinical phenotypes, such as frequent exacerbations and chronic colonization by *Pseudomonas aeruginosa*, supporting the possible role of the *TAS2R38* gene in susceptibility towards respiratory infections. 

However, when we investigated the *TAS2R38* haplotype in 196 COVID-19 patients without PCD [12], no significant associations between the *TAS2R38* haplotype and the presence/severity of COVID-19 were detected.

Our present findings suggest that subjects with PCD may not be at increased risk of COVID-19 and/or of severe outcomes and that the different haplotypes of the gene codifying for the bitter taste receptor TAS2R38 do not seem to correlate with a propensity to SARS-CoV-2 infection. Additionally, the PAV/PAV haplotype does not seem to have a protective role towards SARS-CoV-2 infection.

So far, these data are preliminary and need to be confirmed by studying a greater number of PCD subjects. 

Previous reports [9,10] showed an association between the *TAS2R38* genotype and SARS-CoV-2 infections, mainly in cases of severe symptoms leading to hospitalization and mortality. Our study on adult PCD patients affected by mild to moderate COVID-19 did not identify any significant association; we cannot exclude, however, that an association between *TAS2R38* genotype and infection could be observed in patients with severe COVID-19.

There are several other limitations to our study. First, given the rarity of PCD, we had a relatively small sample size; we performed this investigation mainly on adult patients and, therefore, we could not determine whether the subjects included were representative of the PCD population. We only had self-reported data on test results for SARS-CoV-2 infection, and this may underestimate the overall prevalence of the disease due to unreported asymptomatic cases. The absence of a control group in our study is another possible limitation. Nevertheless, the major strength of this study is that this is the first time that polymorphisms of TAS2R38, in particular, haplotypes PAV/AVI and AVI/AVI, were investigated in relation to a possible greater prevalence or poorer outcome of COVID-19 in PCD patients. This could be a good starting point for future larger studies.

## 4. Materials and Methods

### 4.1. Patient Cohort and Clinical Evaluation

We included PCD patients followed-up at the Centre for Rare Disease of the Pneumology Unit, Policlinico Hospital, in Milan, Italy, and we recorded data from December 2023 to January 2024; most of these patients previously participated in the study on the impact of *TAS2R38* gene polymorphisms on PCD outcome and severity of disease [25]. 

We asked them whether they had contracted COVID-19 in the last four years on the basis of a specific test; what their symptoms, duration of disease, and clinical course were; whether and how many times they had been hospitalized; whether they had been vaccinated against SARS-CoV-2; how many vaccine doses they had received, and whether they experienced any side-effects after vaccination. We also collected information about the lifestyle, physical activity, work environment, and health behaviours such as the use of facemasks. COVID-19 was categorized into no symptoms, mild symptoms (mild fever and/or cough), moderate symptoms (high fever, cough, headache), or severe if they require hospitalization. 

The Ethical Committee of the Hospital approved the protocol and a written informed consent was obtained from all participants or from the parents of pediatric patients. 

### 4.2. Questionnaires

Patients filled the Sino-Nasal Outcome Test-22 (SNOT-22), a questionnaire that is commonly used to document the outcomes of patient-reported chronic rhinosinusitis and is also considered a suitable widely used tool to measure disease-specific quality of life [26]. A copy of SNOT-22 is included as a Appendix A. It rates 22 different symptoms from 0 (no problem) to 5 (problem as bad as it can be) related to rhinological, ear, facial, general, physical, and psychological domains. The scores range from 0 to 110 with high scores indicating greater symptoms. 

Participants who referred having had COVID-19 also filled out a structured questionnaire to evaluate symptoms during the acute phase of infection, called the Acute Respiratory Tract Infection Questionnaire (ARTIQ) [27]. Patients were asked to report COVID-19 symptoms experienced during the acute phase of infection. Specifically, the analyzed symptoms were taste/smell reduction, increased cough, hearing loss, blocked nose, runny nose, sneezing, lacrimation, raucousness, fever, swelling, chills, headache, sore throat, muscle or joint pains, chest pain, sinonasal pain, neck tumefaction, problems of breathing, dyspnea, asthenia, loss of appetite, diarrhea, nausea, vomiting, abdominal pain, dizziness, poor quality of sleep, and difficulty in concentration. The presence of symptoms was registered as a dichotomous variable (1: yes/0: no) and symptom severity was ranked on a 0–2-point scale as none (0), mild (1), and severe (2).

### 4.3. Genetic Analysis (i.e., TAS2R38 Polymorphisms)

This was carried out on DNA extracted from peripheral blood. Briefly, DNA was obtained using the Isohelix extraction protocol-DNA isolation kit (Cell Projects, Kent, UK); genotypes of three *TAS2R38* SNPs (rs1726866, rs713598, and rs10246939) were determined using the TaqMan-probe-based assays (Applied Biosystems, Foster City, CA, USA) [28]. Participants were classified as PAV/AVI heterozygous, PAV/PAV homozygous, and AVI/AVI homozygous. The genetic database relative to *TAS2R38* SNPs was reviewed in light of what has been reported concerning COVID-19.

### 4.4. Statistical Analysis

Demographic and clinical characteristics of participants are described as means with SDs for normally distributed continuous data or as absolute frequency and percentages for categorical data. Differences between the percentages were tested by Fisher's test while those from means were tested by ANOVA analysis; correlation analyses were performed with the Pearson’s test; a logistic binomial regression was applied to study the association between different *TAS2R38* haplotypes and SARS-CoV-2 infection. Statistical significance was estimated for *p* < 0.05. Statistical analysis was performed using R software version 4.1.2 (R Foundation for Statistical Computing, Vienna, Austria).

## Figures and Tables

**Table 1 ijms-25-08635-t001:** Main characteristics of the PCD patients studied (23 F, 17 M).

	Mean ± SD	Patients, N (%)
Age (years)	36.6 ± 16.7	
BMI (kg/m^2^)	18.7 ± 4.6	
Situs viscerum inversus		22 (55%)
Chronic rhinosinusitis		27 (67.5%)
Bronchiectasis		29 (72.5%)
Colonization by *P. aeruginosa*		15 (37.5%)
Bronchial asthma		8 (20%)
Allergy		10 (25%)
SARS-CoV-2 infection		12 (30%)
Duration of symptoms (days)	7.4 ± 4.5	
Severity of infection		
- Mild;		10 (83.3%)
- Moderate;		2 (16.7%)
- Severe.		0 (0%)
Hospitalization		0 (0%)
Vaccine doses	3.5 ± 0.9	
FEV_1_	80.1 ± 17.3	
SNOT-22	28.7 ± 22.2	

**Table 2 ijms-25-08635-t002:** Clinical features of PCD patients COVID-positive in comparison to non-COVID (mean ± SD or number and percentages) and the statistical significance of differences.

	COVID-Acquired (12) 7F, 5M	Non-COVID-Acquired (28) 16F, 12M	Statistical Significance (*p*)
Age (years)	35.9 ± 12.4	37 ± 18.5	0.85
BMI (kg/m^2^)	23.2 ± 3.3	20.1 ± 4.1	0.03 *
Situs viscerum inversus	4 (33.3%)	18 (64.3%)	0.09
Chronic rhinosinusitis	9 (75%)	18 (64.3%)	0.71
Bronchiectasis	10 (83.3%)	19 (67.8%)	0.45
Colonization by *P. aeruginosa*	5 (41.7%)	10 (35.7%)	0.73
Bronchial asthma	2 (16.7%)	6 (21.4%)	1.0
Allergy	2 (16.7%)	8 (28.6%)	0.7
Number of vaccine doses	3.5 ± 0.7	3.4 ± 0.9	0.73
FEV_1_	73.4 ± 18.8	83.1 ± 16.2	0.1
SNOT-22	27.1 ± 20.5	29.9 ± 23.8	0.72

* Statistically significant.

**Table 3 ijms-25-08635-t003:** Clinical features of SARS-CoV-2 infection in PCD patients divided according to *TAS2R38* polymorphisms.

	TAS2R38 Haplotype		
	PAV/PAV (10)	PAV/AVI (17)	AVI/AVI (8)
SARS-CoV-2 infection	3	4	2
Severity of infection			
- Mild;	2	2	2
- Moderate;	1	2	0
- Severe.	0	0	0
Hospitalization	0	0	0
Loss of olfaction/taste	0	0	0
Duration of infection			
- <7 days;	2	3	1
- >7 days.	1	1	1

## Data Availability

The data presented in this study are available on request from the corresponding author.

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
