# Peer review of "TAS2R38 Genotype Does Not Affect SARS-CoV-2 Infection in Primary Ciliary Dyskinesia"

_ijms, 2024, doi:10.3390/ijms25168635_

Round 1

Reviewer 1 Report

Comments and Suggestions for Authors

In the present work Piatti et al conducted a retrospective study on the association between TAS2R38 receptor polymorphisms and SARS-CoV-2 infection in patients with PCD. The study presents several critical issues, including the small number of patients recruited, the absence of Non-PCD controls, the lack of clinical data on the course of SARS-CoV-2 infection. Overall, the study, as presented, is inconsistent and does not significantly move the field forward. Authors should be clarify the main scopus of the paper.

Scientific background issues:

Introduction line 52-53: The role of TAS2Rs against viral infection should be addressed in more depth to explain why it was decided to focus on these genes.

Moreover, what are the possible molecular bases of the involvement of the TAS2R38 receptor in SARS-CoV-2 infection?

Introduction line 64-78: previous works show an association between the TAS2R38 genotype and SARS-CoV-2 infection only in cases of hospitalization and mortality. Moreover, the study of adult patients affected by mild to moderate COVID-19 did not identify any significant association. Basic on this background the present study includes only patients with mild to moderate COVID-19 and did not include PCD patients with severe COVID-19, i.e. the category where it should be most likely to observe an association between TAS2R38 genotype and infection.

Method:

There is no data regarding the SARS-CoV-2 strain contracted by patients.

There are no data regarding patients with PCD who have contracted SARS-CoV-2 in asymptomatic form.

There are no data regarding non-PCD patients as controls.

Since no patient required hospitalization, there are no clinical data on the course of the disease, including the duration of symptoms.

Comments on the Quality of English Language

Minor editing of English language required

Reviewer 2 Report

Comments and Suggestions for Authors

It was with great interest that I read the article several times, the subject addressed by the author is a plus in Covid-19 management correlated with several risk factors. The author has presented a quality piece of work, I just have some small comments to make it even better.

My general comment is that the author could improve the method section, which I find a little expeditious, and Table I could be better organised.

-Line 31 : The author has given the number of female; the number of male should also be mentioned.

- Tab I : It would be a good idea to group similar items together for easier reading: put the mean Sd together and the numbers/percentage together.  It would also be good to name each column

- Line 123 : Asthenia has been used in the text and Tiredness in Table III, it would be good to use the same word in both places, e.g. use Asthenia in the Table III.

- Line 230 : Was informed consent obtained in the same way for paediatric patients? If not, the author should describe this exceptional case.

- Line 232 : The author must show a copy of the questionnaire (even the first page will be sufficient)

- Line 247 : The use of Taq Man probe-based assays must be justified by a reference

Round 2

Reviewer 1 Report

Comments and Suggestions for Authors

The paper is well-written and easy to follow. However, the results are unconvincing, not very new, and ultimately advance the field in little significant ways. I suggest the authors propose the manuscript as a short communication, it would be better in a more virological or clinical journal.

Round 3

Reviewer 1 Report

Comments and Suggestions for Authors

In my opinion, the manuscript is now suitable for publication in IJMS as Communication.